# Quantifying Arm and Leg Movements in 3-Month-Old Infants Using Pose Estimation: Proof of Concept

**DOI:** 10.3390/s24237586

**Published:** 2024-11-27

**Authors:** Marcelo R. Rosales, Janet Simsic, Tondi Kneeland, Jill Heathcock

**Affiliations:** 1School of Health and Rehabilitation Sciences, The Ohio State University, Columbus, OH 43210, USA; 2Heart Center Nationwide Children’s Hospital, Columbus, OH 43210, USA; janet.simsic@osumc.edu; 3School of Health and Rehabilitation Sciences, College of Nursing, The Ohio State University, Columbus, OH 43210, USA; kneeland.12@osu.edu; 4Abigail Wexner Research Institute Nationwide Children’s Hospital, Columbus, OH 43210, USA

**Keywords:** pose estimation, infant, movement, complex congenital heart disease

## Abstract

Background: Pose estimation (PE) has the promise to measure pediatric movement from a video recording. The purpose of this study was to quantify the accuracy of a PE model to detect arm and leg movements in 3-month-old infants with and without (TD, for typical development) complex congenital heart disease (CCHD). Methods: Data from 12 3-month-old infants (N = 6 TD and N = 6 CCHD) were used to assess MediaPipe’s full-body model. Positive predictive value (PPV) and sensitivity assessed the model’s accuracy with behavioral coding. Results: Overall, 499 leg and arm movements were identified, and the model had a PPV of 85% and a sensitivity of 94%. The model’s PPV in TD was 84% and the sensitivity was 93%. The model’s PPV in CCHD was 87% and the sensitivity was 98%. Movements per hour ranged from 399 to 4211 for legs and 236 to 3767 for arms for all participants, similar ranges to the literature on wearables. No group differences were detected. Conclusions: There is a strong promise for PE and models to describe infant movements with accessible and affordable resources—like a cell phone and curated video repositories. These models can be used to further improve developmental assessments of limb function, movement, and changes over time.

## 1. Introduction

Advancements in technology, digital health, and remote monitoring are making the ability to collect sophisticated and objective pediatric movement data feasible, cost-effective, and ecological [1,2]. Devices such as wearable sensors [1,2,3] and portable markerless multi-camera 3D motion analysis systems [4,5,6,7,8] make the collection of human movement data possible in many environments (i.e., home, day care, sports field, research laboratory). The potential of these systems to detect, identify, and measure changes in symptoms, including those that are present in everyday life, is exciting. While these tools produce meaningful results, they require either a device attached to the infant (wearable) or specialized cameras. However, there is another analysis tool that can address these limitations and has shown promise in measuring movements in adults and children from a video recording [9].

This emerging tool that has shown promise for analyzing movements from simple video recordings is pose estimation [6,7,8,9]. Pose estimation can track movement in a video by recording the pixel’s position over time. The pixels from the video measurement can be converted to an x–y–z coordinate for an estimate of position. Depth, or the z-direction, cannot be calculated with a single camera to our knowledge. Cameras project a scene onto a 2D space and data are lost if motion extends outside of the limited field of view. Pose estimation has been used to measure 2D movements of the legs in athletes [10], adults with stroke [9,11,12], and patients with Parkinson’s Disease [9]. Pose estimation has also been used to assess the arm of adults with stroke [9] and the fingers in patients with Parkinson’s Disease [13]. Unlike wearable sensors and markerless motion capture with multiple cameras, pose estimation does not require the participant to wear additional equipment and can be used from a single camera recording for movement primarily in a single plane. This is particularly compelling for infants who are wearing several pieces of medical equipment in intensive care units or who do not want to wear markers for 3D motion analysis [4,5,9]. The potential is high because it means movement and biomechanical data can be collected and analyzed across many environments, with minimal training, at a cost-effective price, part of in-person or remote digital monitoring, and for use with retrospective video repositories and curated clinical data sets [14,15,16,17].

While pose estimation has the promise to quantify pediatric movement from a video recording, firstly, preliminary validation of data processing techniques is required to develop robust measures of infant movement. Most pose estimation models have been developed on adult populations. Infant bodies’ parameters are often difficult to identify with models developed on adult proportions given the anthropometric proportions and size differences (from adults) [18,19]. Additionally, movement type varies greatly across human development, and the methods used to assess adult movement might not be appropriate for pediatric movement. Therefore, the currently available models for pose estimation must be tested and validated for infant use.

Currently, in the pose estimation literature, pediatric movement has been quantified using pixels as the unit of measurement (e.g., pixels per second for velocity, or a change in pixels for displacement) for spontaneous movement of the arms and legs [8,9]. These pixel measurements [6,20] are then correlated with clinical or research tools like the general movement assessment (GMA) or diagnosis of developmental disabilities as an early measure of concurrent or predictive validity. These studies show the potential that pose estimation can (1) estimate and localize body postures with a single camera view accurately, (2) quantify infant movements, (3) be used with automated computer-based methodologies, and (4) have early validity with clinical measures. However, pose estimation can extract meaningful infant behavioral analytics that are easier than pixels to understand for research and clinical practice related to movements [21]. For example, variables such as the frequency counts of movements (i.e., the number of movements or quantity of movements) and how often similar movements are repeating themselves (i.e., the sample entropy or complexity) can be calculated.

Quantity [2,22,23] and complexity [22,24] of movement are powerful measures in the child developmental and rehabilitation literature, because they are correlated with motor skill development and are being developed as a tool for gauging motor rehabilitation of children with developmental disabilities [25,26]. Therefore, there is a need to validate a way to process pose estimation data using units that are commonly used in order to compare and integrate within the current literature base. Quantifying the acceleration of movements in the limbs, similar to how these measures are calculated with wearables, will allow us to use pose estimation to quantify the number of movements and the complexity of movement in infant populations.

The purposes of this study were to (1) test the accuracy of an open-source PE model, called MediaPipe, to detect arm and leg movements in 3-month-old infants with and without complex congenital heart disease (CCHD); (2) compare the frequency and complexity of arm and leg movements between infants with and without CCHD; and (3) establish preliminary concurrent validity with the Test of Infant Motor Performance and the Bayley Scale for Infant and Toddler Development.

Retrospective data from infants with and without CCHD were selected for this proof of concept to assess pose estimation in a clinical population and a comparison group. Infants with CCHD are at risk of neurodevelopmental disabilities and video assessment using pose estimation of spontaneous movements and no manipulation of a sick infant in the pediatric intensive care units might benefit this population. In addition, inclusion of both typical development and CCHD populations were included to make sure there was a sufficient range and variability of spontaneous behaviors.

## 2. Materials and Methods

### 2.1. Participants

A subset of video data from a larger study [27] was used for this study. Participants were full-term infants (>36 weeks gestational age), 3 months of age, with (n = 6) and without (n = 6) complex congenital heart disease (CCHD). All the infants with CCHD required surgical intervention within the first months of life.

### 2.2. Procedures

Participants were assessed in their homes using the Test of Infant Motor Performance (TIMP) [28], the Bayley Scale for Infant and Toddler Development (3rd edition) [29], and a 15 min contingency mobile paradigm. In the contingency paradigm, movements of the right leg activate the mobile. Video data from the contingency paradigm were used in this study and were recorded by a Sony 8mm CCD-TRV608 at 30 frames per second. The whole 15 min video was processed using the PE model.

Videos were selected if, for the entire video, the infant was in full view, in the frontal plane, and the mobile did not come into the camera view. This resulted in videos that could view the baby for the whole paradigm and did not have time points where external entities (i.e., the mobile, a parent, or an assessor) blocked the view of the child, which might interfere with locating virtual anatomical markers. The video repository contained 46 videos (35 infants with CCHD and 11 infants with typical development (TD)) and 12 were used. Five videos of infants with TD were excluded for occlusions of the infant (i.e., the infant was not in full camera view for the entire video), yielding six videos of infants with TD. The video files of infants with CCHD were randomly run through the model until a balanced group was acquired. Thirteen video files were needed to acquire a sample of six video files from the infants with CCHD. Seven video files from the infants with CCHD were excluded for occlusions of the infant.

### 2.3. Data Processing

Video data (30 frames per second) were imported and processed using custom python (version 3.9) and MATLAB (R2023b) scripts. Python (version 3.9) was used to extract the position in the x and y coordinates for 33 anatomical markers (referred to as virtual markers) from the MediaPipe model (Figure 1) [30]. MATLAB (2023b) was used to quantify the acceleration of arm and leg movement using the position of the virtual markers of the ankle and wrist. Arm and leg movements are the focus of this manuscript.

Prior to MATLAB processing, output videos were visually inspected to determine that the pose estimation model was correctly placing the virtual markers on the correct anatomical locations of an infant. Unrealistic values for virtual markers can be caused by items and people that block the view of the child, such as the rail of a crib, a caregiver’s hand, or part of the mobile coming into camera view. Obstructions that only occurred for a second of the video data (i.e., an ID card for labeling the video data) or had infrequent occurrences (i.e., a caregiver manipulating a pacifier for 1 s 2 times during the collection) were included in the data set. Video data with obstructions that occurred for longer than 10 s (a crib rail, obstructing the view of a child) or frequently occurred (i.e., a mobile consistently coming into view) were excluded. Infant behaviors like bringing hands to midline, placing hands in the mouth, and grasping feet did not interfere with the model’s trace of the infant and were included in the data set.

Since MediaPipe exports the virtual markers in pixels, we converted pixels to meters using the individual x and y coordinates for both virtual markers of the shoulder, a standardized norm value for shoulder-to-shoulder distance (S2S) for 3-month-old infants (0.18415 m) [31], and trigonometry. The distance between the virtual shoulder markers was calculated by taking the arc tangent of the absolute difference between the y coordinates divided by the absolute difference between the x coordinates (Equation (1)). This was performed for each frame of data, resulting in the shoulder-to-shoulder distance in pixels for every frame. The median value for shoulder-to-shoulder distance was quantified for the whole file and then was divided by 0.18415 m (the norm value for shoulder-to-shoulder distance in 3-month-old infants) (Equation (2)). This resulted in the conversion of meters to pixels and was used to convert pixels to meters for all virtual markers in each participant’s data set. These data were smoothed using the Gaussian filter in MATLAB with a window frame of 8.
(1)$$S2S=\tan^{-1}\left(\frac{\left|y_{li}-y_{ri}\right|}{\left|x_{li}-x_{ri}\right|}\right)$$
(2)$$Conversion=\frac{Median\;S2S}{0.18415}$$

Position data from the virtual ankle and wrist markers and time were used to quantify acceleration of the limbs (i.e., the change in velocity over time in *x* and *y*). Infant leg and arm movements were quantified using the peak movement acceleration of the ankle and wrist virtual markers for both sides. This was adapted from Smith et al. (2015) [2]. Acceleration was quantified using the square root for the sum of squares from the acceleration in the *x* and *y* coordinate of each leg and arm (see Formula (3)). Acceleration peaks were identified using the findpeaks function in MATLAB. Each peak was identified using the video timeline that was exported from the Python (version 3.9) output using the MediaPipe model (see Figure 2). Arm and leg movements were defined as an acceleration peak greater than 3 m/s^2^ and occurring in more than 10 frames of video data (i.e., 0.33 s) from the prior movement of the corresponding limb. The acceleration threshold was chosen based on Fitter et al. (2019) [32] and the time threshold of 10 frames of video data (i.e., 0.33 s) was chosen based on piloting.
(3)$$acceleration=\sqrt{x^{2}\;+\;y^{2}}$$

Leg and arm movement rates per hour were calculated by dividing the number of movements produced in the whole 15 min video and the duration of the video in minutes. Then, the results were multiplied by 60 min to convert the value to movements per hour. This unit was chosen to more easily compare the results from pose estimation data to previously reported values in the wearable sensor literature, which reports movements per hour [2,25].

Using the acceleration signal from each limb, sample entropy for each limb was calculated in MATLAB (2023b) using a custom program. This program has been used in Deng et al. (2022) [22] and Smith et al. (2017) [24] and was developed based off Richman and Moormman (2000) [33].

### 2.4. Behavioral Coding of Leg and Arm Movements

Firstly, behavioral coding was used as the gold standard and identified leg and arm movement for 20 s for each video as a measure of validity [2]. A movement expert with doctoral training identified the onset and offset of a leg or arm movement. A leg movement was operationally defined as a visible change in the ankle’s location (i.e., displacement) with flexion or extension of the hip and knee. An arm movement was operationally defined as a visible displacement of the wrist with joint movement at the shoulder or elbow. The timing of movements from the annotated video file was compared with results from the MATLAB code and the agreements and disagreements between the data were identified (Table 1).

### 2.5. Measurements

The Bayley Scale for Infant and Toddler Development [29] is an assessment that measures a child’s cognitive, language, and motor development. Children perform a set of items for each domain until they score a 0 on 5 consecutive items. This assessment is used in children 0–42 months of age. Bayley scores for the cognitive, communication, and motor domains were used for this analysis.

The Test of Infant Motor Performance (TIMP) [28] is an assessment of an infant’s motor development. It is used in infants 34 weeks post-conception to 4 months post-term and was developed to identify infants with delayed motor development and to track the progress of children with typical development.

### 2.6. Statistics

All statistics were performed in MATLAB (2023b). The positive predictive value (i.e., the likelihood of identifying a movement correctly) and sensitivity (i.e., it is the likelihood that a movement is identified, if a movement is known to have occurred) were used to assess the accuracy of the model. The positive predictive value was calculated by dividing the total number of agreed movements identified between behavioral coding and pose estimation by the sum of movement identified by both methods. Sensitivity was calculated by dividing the total number of agreed movements identified by both methods by the total number of movements identified using behavioral coding.

Wilcoxon rank sum tests were used to determine differences between CCHD and infants with typical development (TD) for the positive predictive value, limb movement rate, sample entropy, and Bayley scores and TIMP scores. Pearson correlations were used to determine effect sizes for the comparison of CCHD and TD for the following variables: average leg and arm movement rate (the average of left and right limb) and sample entropy for all limbs. Pearson r can be explained using the following values for r: small effect was |r| > 0.1, moderate effect |r| > 0.3, and large effect |r| > 0.5 [34]. Spearman correlations were used to examine the association between the average leg and arm movement rate, sample entropy, and Bayley and TIMP scores for concurrent validity. Pearson r was also used to assess the effect size between the movement rate of the arms and legs, with the Bayley and TIMP scores.

## 3. Results

### 3.1. Participants

The median age for TD was 3.54 months (range: 3.4–3.9) and CCHD was 3.4 months (range: 3.06–3.73). A total of 75% were male (n = 3 female) and racial composition of the participants was 83% White and 16% Black. None of the children were Hispanic.

### 3.2. Model’s Accuracy to Detect Leg and Arm Movement in TD and CCHD

We identified 499 leg and arm movements, and the pose estimation model had a positive predictive value of 85% and a sensitivity 94% (Table 1a). The pose estimation model’s positive predictive value in TD was 84% and sensitivity was 93% (Table 1b). The pose estimation model’s positive predictive value in CCHD was 87% and the sensitivity was 98%. (Table 1c). No significant differences were found between the groups for the pose estimation model’s positive predictive value (*p* > 0.05).

**Table 1 sensors-24-07586-t001:** Two-by-two contingency tables comparing the agreement and disagreements between the detection of limb movements using the pose estimation and behavioral coding for the following: (**a**) the combination of all children, (**b**) children with typical development, and (**c**) children with complex congenital heart disease. The section with no movement (i.e., the cell with “No behavioral coded movement” and “No pose estimation Movement”) is left blank because it is not possible to calculate the number of movements that did not happen.

(**a**)		
	Behavioral coded movement	No behavioral coded movement
Pose estimation movement	499	57
No pose estimation movement	30	
(**b**)		
	Behavioral coded movement	No behavioral coded movement
Pose estimation movement	345	37
No pose estimation movement	27	
(**c**)		
	Behavioral coded movement	No behavioral coded movement
Pose estimation movement	154	20
No pose estimation movement	3	

When examining leg and arm movements separately, overall, the positive predictive value was 84% in legs (Table 2a) and 86% in arms (Table 3a). Overall, the sensitivity was 94% in legs (Table 2a) and 95% in arms (Table 3a). For TD, the positive predictive value was 83% in legs (Table 2b) and 86% in arms (Table 3b), while the sensitivity was 92% in legs (Table 2b) and 94% in arms (Table 3b). For CCHD, the positive predictive value was 87% in legs (Table 2c) and 87% in arms (Table 3c), while the sensitivity was 99% in legs (Table 2c) and 97% in arms (Table 3c).

### 3.3. Differences in Average Limb Movement Rate

For individual infants using movements/hour, movement rates ranged from 399 to 4211 movements per hour for legs and 236 to 3767 movements per hour for arms. The median movement rate for TD was 2357 movements per hour for legs (range: 1327–4211 movements per hour) and 1373 movements per hour for arms (range: 912–3767 movements per hour). The median movement rate for CCHD was 1199 movements per hour for legs (range: 399–4155 movements per hour) and 1231 movements per hour for arms (range: 236–3764 movements per hour). TD tended to produce more movements than infants with CCDH; however, no significant differences were found for the hourly leg and arm movement rates (Figure 3). When comparing groups, there was a medium effect size for the leg rate (r = 0.34) and a small effect size for the arm rate (r = 0.25). This suggests that compared to CCHD, the leg rate and arm rate are higher in the TD group.

### 3.4. Sample Entropy of the Acceleration Signal for Arm and Leg Movements

Median sample entropy in the left and right arms was 0.27 and 0.32 for CCHD, respectively, and 0.29 and 0.27 for TD. Median sample entropy in the left and right legs was 0.25 and 0.32 for CCHD, respectively, and 0.37 and 0.41 for TD. The box plot in Figure 4 depicts the sample entropy for both groups in all four limbs. There was no statistical difference between the groups for sample entropy for left and right arm and leg movement (*p* > 0.05). Infants with TD tended to have larger sample entropies for both side and limbs, meaning that the TD group may have produced more variability in their movements. There was a medium effect size for left leg movements (r = 0.32) and a low effect size for left (r = 0.25) and right (r = 0.13) arm movements and right leg movements (r = 0.15), suggesting that the TD group showed more variability in the leg that activated the mobile (i.e., the right leg).

### 3.5. Associations with Movement Rate and Developmental Assessments

Bayley communication was significantly correlated with the arm movement rate (*p* = 0.04, r = 0.6). No other associations were found between the leg and arm movement rate and the TIMP and the motor and cognitive scale (*p* > 0.05) (see Table 4). However, Pearson’s correlations showed medium effect sizes between TIMP scores and the leg (r = 0.30) and arm (r = 0.38) movement rate, as well as Bayley communication scores and arm movement rate (r = 0.35). This suggests that infants with higher TIMP scores had higher leg and arm movement rates, and those with higher Bayley communication scores had higher arm movement rates. Small effect sizes were found between Bayley cognitive scores and arm movement rate (r = 0.25), and Bayley communication scores and leg movement rate (r = 0.23). This suggests that infants with higher Bayley cognitive scores had higher arm movement rates, and those with higher Bayley communication scores had higher leg movement rates.

## 4. Discussion

This paper preliminarily validates a data processing technique to quantify and describe infant limb movement. The two groups were compared to (1) validate these measures in a clinical population as well as a population with typical development (TD), and (2) perform an exploratory comparison that will lead to further robust questions using pose estimation. The methodology presented in this study shows that there is considerable promise for quantifying movement in 3-month-old infants using pose estimation. Firstly, there was high PVV (i.e., all values above 80%) and sensitivity (i.e., all values about 90%) with behavioral coding. Secondly, using the model alone, the ranges of arm and leg movements are similar to those reported in the wearables literature. Thirdly, although nonsignificant hypothesis testing was performed to compare CCHD to TD, the following reasons support the trends and effect sizes observed: (1) the right leg showed higher values as would be expected in the mobile paradigm with a tethered right leg, (2) the CCHD infants had lower mean rates as might be expected by an infant population that spent considerable time in the PICU and had major heart surgery, and (3) the complexity measured by sample entropy is similar to those values reported in the literature for infant movements using other methods.

By developing a method for integrating a familiar measurement into the system, pixels can be converted to meters and be used to measure movement in standardized ways that are easily understood by clinicians and researchers. Additionally, the data processing techniques described in the study show that we can also quantify the complexity of movement that the infants are producing through measurement of the sample entropy. Therefore, it is possible to quantify the amount of movement and its complexity through video data. The potential to analyze movements from any video recording, through digital health appointments, during clinical assessments, and with any video repository may add a layer of precision and prediction that was once only possible in a laboratory setting.

### 4.1. Frequency

Quantifying common variables used in the wearable sensor literature and applying it specifically to pose estimation data with pixels provides a way to study infant movement that is translatable to other areas of the biomechanical literature. The movement rate values in the two groups were very similar to those that have been previously reported in the wearable sensor literature and behavioral coding literature. The hourly leg movement rate has ranged from 813 to 3671 in infants with TD [2], and in recent studies, it has been found to be higher [26]. For the arm movement rate data, 3-month-old infants perform about 2011 movements per hour [35]. These values are similar to the data reported in this paper (leg rate range: 399–4211 movements per hour; arm rate range: 236–3767 movements per hour) (see Figure 3).

In addition to the movement rate values being similar to values reported in the wearable sensor literature, the trend for infants with TD to produce more movements than CCHD is represented in the literature. Children at risk of neurodevelopmental disabilities, including CCHD, have been found to produce fewer movements compared to those with TD [25,36,37,38]. It is for this reason that early interventions sometimes focus on increasing the movement rate and motor experience as soon as possible [38,39,40]. The results from this paper offer a measurement that could be implemented in a variety of settings to help gauge the amount of movement infants at risk of neurodevelopmental disabilities are producing. Additionally, future work could explore the possibility of using pose estimation in combination with other tools as a biomarker to recommend early intervention and treatment response.

The sample entropy values of arm and leg movements in our sample were slightly higher than those reported in the literature, but still within the range of the reported values [22,24]. The movement characteristics of sample entropy are difficult to compare between studies (e.g., different movements, tasks, and participant characteristics) because the interpretation of sample entropy is context-specific. For a healthy adult walking on a treadmill, you would expect a low sample entropy, because their movements will be consistent and less variable, while an infant learning to walk would have a higher sample entropy, due to more variability in their movements. Therefore, comparisons between the sample entropy found in this study with past studies should be examined with caution.

While wearable sensors can be used to collect full data recordings and can account for movements that are the results of rotation [2,41], pose estimation data can be used to study concepts that are challenging using wearable sensor methodology. For example, full-day wearable sensor data do not describe the context in which movement is occurring unless it is paired with written diaries (e.g., activity logs) or surveys (e.g., ecological momentary assessment) [1]. Pose estimation offers the ability to look at a recording of the movement and further analyze it once the data have been processed. This offers the potential to study sensorimotor experience and how infants produce motor behaviors in response to different stimuli like touch from a caregiver or themselves, voice, and light. For these studies, pose estimation may support or make behavioral coding efficient for movement variables, making data processing much quicker and more feasible with larger samples.

### 4.2. Two-Dimensional vs. Three-Dimensional Motion Analysis

In comparison to markered 3D motion analysis [42], pose estimation does not require an infant to wear additional equipment like markers, and thus can be performed at a much lower cost [9]. This is very promising because pose estimation could be used in infant populations (or any participants) that do not tolerate markers or pull off markers when they are placed on landmarks. While pose estimation is limited by the fact that all the anatomical landmarks in the model need to be present on the participant in the video recording, future work can focus on creating new pose estimation models that can analyze infant movement in the prone position, when a specific body part is occluded due to therapies like constrain-induced movement therapy [43], or in a specific body position like prone or sitting.

### 4.3. Preliminary Findings Associating Movement Rate and Clinical Assessments

While our sample size lacked the power to detect associations between the movement rate and clinical assessment data, the results suggest statistical trends that inform further study. We found several medium effects when looking at the association between the leg and arm movement rate and the TIMP, as well as the Bayley communication score and arm movement rate. This suggests possible associations between movement captured using pose estimation data and clinical tools used to evaluate and gauge a patient’s motor improvements during and after a treatment. These associations should be explored further with a larger sample in the attempts to look at future biomarkers that could identify developmental disabilities at earlier ages.

### 4.4. Limitations

The sample size in this study was determined by the amount of data available according to the specific parameters set to obtain an accurate trace of the infant’s movement. This may be a contributing factor for not finding a significant difference between the two groups. However, our effect sizes show promising results and suggest that these data can be used to determine movement differences in these two populations with a larger sample size. Future studies with a larger sample could examine movement differences in different pediatric populations (i.e., children at risk of neurodevelopmental delays) that may help identify predictors of optimal developmental outcomes. One direction is to use pose estimation in the neonatal intensive care unit. Pose estimation could be used to examine strategies to promote optimal neuromotor outcomes and to better detect developmental disabilities such as cerebral palsy.

Another limitation is that the amount of data from the video files could have been maximized using video trimming or cropping the video file to focus on the region of interest. However, these data were not collected for pose estimation and were used respectively. Future work should aim to present the child in the camera view for the full length of the video, or design projects that allow for segmenting data into blocks.

The sampling rate for the video data was 30 frames per second. Chambers et al. (2020) [8] suggest that pose estimation data should be collected at the highest sampling rate possible with appropriate lighting to improve accuracy. Higher sampling rates are possible given the rapid advancement in the accessibility of video camera data. In fact, most Apple products (e.g., iPhone and iPad) can record video data at 60 frames per second. While higher sampling rates will most likely result in more accurate data, future study is needed to examine how much more accurate these data are and if the increase in accuracy outweighs the cost for storing twice as much data.

## 5. Conclusions

These novel data and analysis show that we can quantify the amount and complexity of infant limb movements using pose estimation. Since pose estimation only requires a single video camera, future studies aimed at understanding the neural correlates of infant movement could be conducted in lower-income areas of the world, similar to the wearables sensor [44] and smartphone literature [1,45], and in situations where devices like wearable sensors cannot be placed on a child. Thus, pose estimation opens new and inexpensive avenues for the analysis of infant movement and the early detection of developmental disabilities.

## Figures and Tables

**Figure 1 sensors-24-07586-f001:**
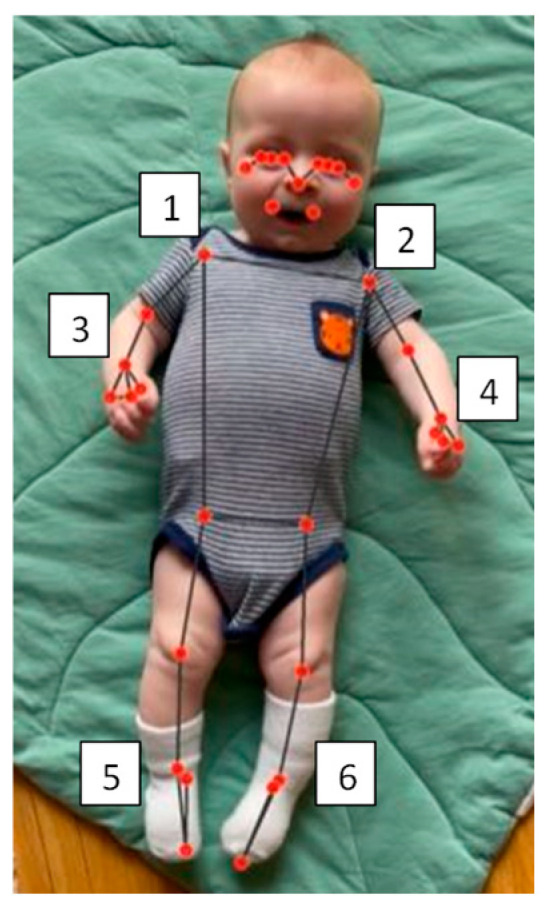
Sample image of all 33 anatomical landmarks on a child. The added number indicates the following virtual markers used for the study: (1) right shoulder, (2) left shoulder, (3) right wrist, (4) left wrist, (5) right ankle, and (6) left ankle.

**Figure 2 sensors-24-07586-f002:**
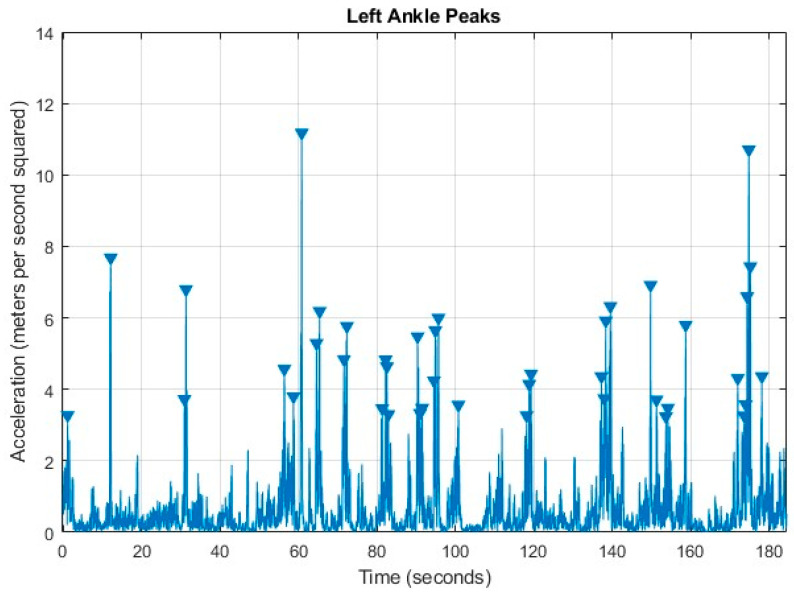
Example of movement count for a single limb. In this case, we display the movement of the left ankle. Triangles represent each occurrence of a movement.

**Figure 3 sensors-24-07586-f003:**
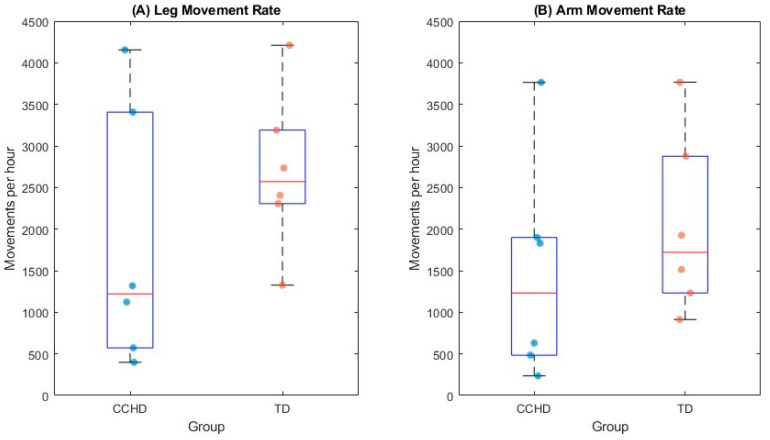
Box plots for leg (**A**) and arm (**B**) movement rate between infants with (CCHD) and without (TD) complex congenital heart disease. Dots on each plot represent average individual movement rate data. Edges of the box represent the 25th and 75th percentile, the middle line in the box is the median, and whiskers represent maximum and minimum values.

**Figure 4 sensors-24-07586-f004:**
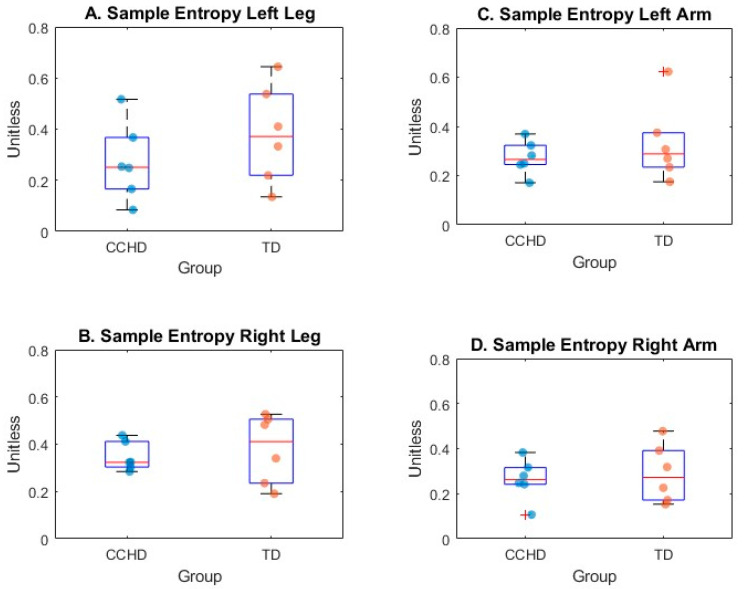
Box plots of sample entropy for leg and arm movements for infants with (CCHD) and without (TD) complex congenital heart disease. The side and limb for each plot is as follows: (**A**) left leg, (**B**) right leg, (**C**) left arm, (**D**) right arm. Dots on each plot represent individual data. Edges of the box represent the 25th and 75th percentile, the middle line in the box is the median, and whiskers represent maximum and minimum values. A plus represents a potential outlier according to the MATLAB function.

**Table 2 sensors-24-07586-t002:** Two-by-two contingency tables comparing the agreement and disagreements between the detection of leg movements using the pose estimation and behavioral coding for the following: (**a**) the combination of all children, (**b**) children with typical development, and (**c**) children with complex congenital heart disease. The section with no movement (i.e., the cell with “No behavioral coded movement” and “No pose estimation Movement”) is left blank because it is not possible to calculate the number of movements that did not happen.

(**a**)		
	Behavioral coded movement	No behavioral coded movement
Pose estimation movement	255	31
No pose estimation movement	16	
(**b**)		
	Behavioral coded movement	No behavioral coded movement
Pose estimation movement	162	18
No pose estimation movement	15	
(**c**)		
	Behavioral coded movement	No behavioral coded movement
Pose estimation movement	93	13
No pose estimation movement	1	

**Table 3 sensors-24-07586-t003:** Two-by-two contingency tables comparing the agreement and disagreements between the detection of arm movements using the pose estimation and behavioral coding for the following: (**a**) the combination of all children, (**b**) children with typical development, and (**c**) children with complex congenital heart disease. The section with no movement (i.e., the cell with “No behavioral coded movement” and “No pose estimation Movement”) is left blank because it is not possible to calculate the number of movements that did not happen.

(**a**)		
	Behavioral coded movement	No behavioral coded movement
Pose estimation movement	244	26
No pose estimation movement	14	
(**b**)		
	Behavioral coded movement	No behavioral coded movement
Pose estimation movement	183	19
No pose estimation movement	12	
(**c**)		
	Behavioral coded movement	No behavioral coded movement
Pose estimation movement	61	7
No pose estimation movement	2	

**Table 4 sensors-24-07586-t004:** Spearman correlation results when comparing average leg and arm movement rate with the Bayley Scale for Infant and Toddler Development’s motor scale and Test for Infant Motor Performance (TIMP).

Movement Variable	Developmental Assessment	r	*p*
Leg movement rate	Bayley—Cognitive	r = −0.15	*p* = 0.64
	Bayley—Communication	r = 0.33	*p* = 0.29
	Bayley—Motor	r = −0.01	*p* = 0.97
	TIMP	r = 0.47	*p* = 0.12
Arm movement rate	Bayley—Cognitive	r = −0.34	*p* = 0.28
	Bayley—Communication	r = 0.60	*p* = 0.04
	Bayley—Motor	r = 0.10	*p* = 0.76
	TIMP	r = 0.23	*p* = 0.35

## Data Availability

Data will be made available upon request from the corresponding author.

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
