# Peer review of "Quantifying Arm and Leg Movements in 3-Month-Old Infants Using Pose Estimation: Proof of Concept"

_sensors, 2024, doi:10.3390/s24237586_

Round 1
Reviewer 1 Report
Comments and Suggestions for Authors
This is an interesting application of Pose Estimation that has the potential to impact rehabilitation research and practice. Overall, this manuscript is well done and easy to read; however, I have a few major and minor comments to improve the impact and clarity.
Major:
1. Please include a justification for why CCHD was used as patient population and why it might be ideal at this stage of work. Additionally, a discussion about this application of this technology in other populations in future work would be useful.
2. There are several reasons videos were excluded (lines 110-114 and lines 130-134). It would be useful to know how many videos were excluded based on this and how many were analyzed. Also, it would be valuable to know how well PE worked in these ‘messy’ videos. What is the rationale for excluding these when the potential power of PE is applying it to everyday videos, which may include obstacles. Could you do some sort of sub analysis with these videos?
3. Related to comment 2, it is unclear how much video was analyzed. Given that you are creating a movement/hour measurement it would be helpful to know how many hours per infant was analyzed.
4. PPV and sensitivity are only presented for combined UE and LE movements, but other metrics are presented for UE and LE separately. It would be very useful to know if the PPV and sensitivity is the same for UE and LE.
Minor:
1. It was initially lost on me that quantifying movement literally meant movement counts. It would be helpful to define quantity as counts earlier in the introduction.
2. Line 89 seems to have a typo related to quantify and quantity.
3. Purpose 2 (i.e., comparing the frequency/complexity of movement between groups) seems a little out of place in this validation work. I think explaining a bit why this contributes to establishing the validity would be helpful. This is done fairly well in line 320-326 of the discussion but should be done earlier.
4. Purpose 3 on line 85 only addresses TIMP but the Bayley is also examined.
5. I suggest that Table 2 be cleaned up a bit. For example, I don’t think there needs to be 2 sets of boxes for leg/arm movement rate and there should be separate columns for the spearman correlation and the p value.
6. I believe that Figures 3 and 4 are individual level averages but I’m not certain. Please clarify this in the text and legends. Also given that your sample is small, I would suggest showing individual data on the figures.

Author Response
Please see attachment in reviewer 1 section.

Reviewer 2 Report
Comments and Suggestions for Authors Please consider providing some more specific comments addressing the following points: * What is the main question addressed by the research? Overall, this very well written manuscript reports a method to quantify the amount and complexity of limb movements in enfants using pose estimations. The purpose of this study was to quantify the accuracy of a pose estimation (PE) model to detect arm and leg movements in 3-month-old infants with and without (TD, for typical development) complex congenital heart disease (CCHD). * Do you consider the topic original or relevant to the field? Does it address a specific gap in the field? Please also explain why this is/ is not the case. Most pose estimation models have been developed on adult populations. Because infant bodies' parameters are often difficult to identify with models developed on adult proportions given the anthropometric proportions and sizes differences. While pose estimation has the promise to quantify pediatric movement from a video recording, first, preliminary validation of data processing techniques is required to develop robust measures of infant movement. * What does it add to the subject area compared with other published material? Leg and arm movements were identified by the pose estimation, and the model had a positive predictive value of 88% and a sensitivity 96%. The pose estimation model's positive predictive value in TD was 88% and sensitivity was 95%. The pose estimation model's positive predictive value in CCHD was 87% and the sensitivity was 98%. This paper preliminarily validates a data processing technique to quantify and describe infant limb movements. The methodology presented in this study shows that there is considerable promise for quantifying movement in 3-month-old infants using pose estimation. * What specific improvements should the authors consider regarding the methodology? What further controls should be considered? Authors should address comment below: Lines 44-46 To my knowledge, there is no way to measure 3-D from a single 2-D camera. Therefore, please consider re-writing lines 44-46 to say Pose Estimation can measure 2-D motion or provide a reference that demonstrate pose estimation can measure 3-D motions. (See page three paragraph two of Stenum, J.; Cherry-Allen, K.M.; Pyles, C.O.; Reetzke, R.D.; Vignos, M.F.; Roemmich, R.T. Applications of Pose Estimation in Human Health and Performance across the Lifespan. Sensors 2021, 21, 7315. https://doi.org/10.3390/ s21217315) Small sample sizes. Replicate with larger sample size. Also, perform similar study with infants at risk of having developmental and or cognitive delays. * Are the conclusions consistent with the evidence and arguments presented and do they address the main question posed? Conclusions are appropriate. Pose estimations have the potential to analyze movements of infants using videos. * Are the references appropriate? Yes * Any additional comments on the tables and figures. Tables and figures are concise and easy to read.
Author Response
Please see the attachment in reviewer 2 section.

Round 2
Reviewer 1 Report
Comments and Suggestions for Authors
My comments were mostly addressed. One last thing I would like addressed is the number of videos included/excluded. I understand that details about which videos can/cannot be analyzed is beyond the scope of this work; however, I still think providing the number of videos that were excluded would improve the transparency of the work [i.e., how many children did you have video from in the original study, how many were included (you have this), and how many were excluded]. One sentence (probably around lines 118-122) would be sufficient.
Author Response
- My comments were mostly addressed. One last thing I would like addressed is the number of videos included/excluded. I understand that details about which videos can/cannot be analyzed is beyond the scope of this work; however, I still think providing the number of videos that were excluded would improve the transparency of the work [i.e., how many children did you have video from in the original study, how many were included (you have this), and how many were excluded]. One sentence (probably around lines 118-122) would be sufficient.
- Response: These data were added to the manuscript in the section recommended. The section reads: “The video repository contained 46 videos (35 infants with CCHD and 11 infants with typical development (TD)) and 12 were used. Five videos of infants with TD were excluded for occlusions of the infant (i.e. the infants was not in full camera view for the entire video), yielding 6 videos of infants with TD. The video files of infants with CCHD were randomly run through the model until a balanced group was acquired. Thirteen video files were needed to acquire a sample of 6 video files from the infants with CCHD. Seven video files from the infants with CCHD were excluded for occlusions of the infant.”
